# Peer Support and Overdose Prevention Responses: A Systematic ‘State-of-the-Art’ Review

**DOI:** 10.3390/ijerph182212073

**Published:** 2021-11-17

**Authors:** Fiona Mercer, Joanna Astrid Miler, Bernie Pauly, Hannah Carver, Kristina Hnízdilová, Rebecca Foster, Tessa Parkes

**Affiliations:** 1Salvation Army Centre for Addiction Services and Research, Faculty of Social Sciences, University of Stirling, Stirling FK9 4LA, Scotland, UK; joanna.miler@stir.ac.uk (J.A.M.); hannah.carver@stir.ac.uk (H.C.); kristina.hnizdilova@stir.ac.uk (K.H.); rebecca.foster@stir.ac.uk (R.F.); t.s.parkes@stir.ac.uk (T.P.); 2Canadian Institute for Substance Use Research, University of Victoria, Victoria, BC V8P 5C2, Canada; bpauly@uvic.ca

**Keywords:** harm reduction, peer support, peer-involved interventions, illicit drug use, overdose prevention, lived experience, systematic review, state-of-the art review

## Abstract

Overdose prevention for people who use illicit drugs is essential during the current overdose crisis. Peer support is a process whereby individuals with lived or living experience of a particular phenomenon provide support to others by explicitly drawing on these experiences. This review provides a systematic search and evidence synthesis of peer support within overdose prevention interventions for people who use illicit drugs. A systematic search of six databases (CINAHL, SocINDEX, PsycINFO, MEDLINE, Scopus, and Web of Knowledge) was conducted in November 2020 for papers published in English between 2000 and 2020. Following screening and full-text review, 46 papers met criteria and were included in this review. A thematic analysis approach was used to synthesize themes. Important findings include: the value of peers in creating trusted services; the diversity of peers’ roles; the implications of barriers on peer-involved overdose prevention interventions; and the stress and trauma experienced by peers. Peers play a pivotal role in overdose prevention interventions for people who use illicit drugs and are essential to the acceptability and feasibility of such services. However, peers face considerable challenges within their roles, including trauma and burnout. Future interventions must consider how to support and strengthen peer roles in overdose settings.

## 1. Introduction

Drug-related overdose is a public health crisis in many European countries [1] and North America [2]. Drug-related death (DRD) is a leading cause of death and is now one of the main causes of the recent decrease in life expectancy in the United States and Canada [3,4,5]. In Europe, illicit and prescribed opioids, including methadone and buprenorphine for opioid substitution treatment (OST), are implicated in around 80–90% of DRDs [6]. The high number of preventable DRDs are related to many factors, including, but not limited to: polysubstance use and an increasing use of stimulants in addition to opioids; new psychoactive substances such as illicit benzodiazepines; and the introduction of synthetic opioids into the illicit opioid supply [6,7]. Those at high risk of overdoses are individuals who are structurally vulnerable and face numerous additional health and social problems, including unstable housing [8], poverty or financial instability [9], poor physical and/or mental ill health [10], and criminalization [11].

Harm reduction strategies are effective in reducing morbidity and mortality in people who use illicit drugs [12]. Harm reduction strategies to reduce overdoses include: safer use education; supervised consumption sites (SCS); overdose prevention sites (OPS); naloxone supply and administration; OST; drug checking; and peer support [13]. To effectively address the overdose crisis, the response must be multi-faceted and prompt, in addition to tackling the underlying social and psychological causes of the crisis [7]. Harm reduction is focused on preventing the harms of drug use, not drug use per se, and is grounded in health, social justice, and human rights perspectives to “minimize negative health social and legal impacts associated with drug use, drug policies, and drug laws” [14]. Harm reduction is rooted in four principles which commit to: respecting the rights of people who use drugs (PWUD), along with their communities and families, and treating all with dignity and respect; engaging with and reporting high-quality evidence from harm reduction research which is safe, effective, and feasible; social justice and meaningful collaboration with PWUD and their networks to minimize health and social exclusion and address systemic inequalities; and the avoidance of stigma and meeting people where they are at [14].

PWUD are at risk of drug-related death and other health harms [15], and there have been significant efforts by PWUD themselves to reduce harms. Indeed, there is a long history of PWUD organizing, developing, and delivering harm reduction resources and services in the community [16]. To protect the health and safety of their communities, PWUDs have implemented a range of harm reduction practices, such as the distribution of sterile syringes by activists and front-line workers. Through the 1980s and 1990s, harm reduction became institutionalized in many settings, yet in countries with more conservative laws it remains politicized and reliant on a degree of civil disobedience [17]. Grassroots unsanctioned practices and spaces play an essential role in trying to mitigate the current overdose crisis [18]. At a community level, harm reduction for PWUD also extends to include a greater emphasis on agency and community building [19,20].

In a previous systematic review of the roles of people who inject drugs in harm reduction initiatives, Marshall et al. [16] identify people with lived or living experience of using drugs as being central to the development of harm reduction interventions. The authors identify that peers deliver harm reduction services in a way which is a unique product of their own experiences [16]. PWUD are essential in effective and client centred services, policies, and programs [21,22,23]. Peers lead in a meaningful way, reducing both harms associated with illicit drug use and related structural violence. Although peers play an essential role in harm reduction interventions, they face considerable stress in these roles which mirrors the societal stigmatization of drug use [24]. Whilst exposure to even a single fatal or non-fatal overdose has been linked to stress, trauma, burnout, and compassion fatigue [25], there is limited support for peers and little recognition of the systemic stressors they face in harm reduction work [24]. The authors emphasize that, despite the complex and demanding nature of the work, peers often feel undervalued due to a lack of organizational understanding regarding their roles and the value of experiential knowledge [24,25].

While not all overdoses are fatal [1], the actions of PWUD in overdose situations are vital to reversing overdose [8,26,27]. Research has highlighted a moral ‘code of conduct’ amongst PWUD which includes assisting other PWUD in the event of an overdose [28,29]. This is when peers enable harm reduction through calling emergency services, administering naloxone, or providing cardiopulmonary resuscitation (CPR) [8,26,27,28,30,31]. Whilst peers are not solely responsible for harm reduction, expansion of peer involvement within harm reduction activities could actively reduce a wide range of harms [32]. Such involvement also has numerous benefits for peers, such as increased feelings of confidence [26] and empowerment [33]. In addition to building ties with the wider community, peer involvement increases intervention feasibility, acceptability, and overall quality of services provided [32]. However, unfair or inequitable compensation, potential devaluation and exploitation, and trauma and grief experienced at work can lead to peer worker burnout [32].

Despite a wide range of research on peer support within overdose prevention, there has been no attempt to synthesize findings to provide general lessons learned and inform further development of the field in relation to recommendations for policy, practice, and research. This article is a ‘state-of-the-art’ review, providing a systematic search and evidence synthesis of relevant literature to address this gap.

## 2. Materials and Methods

### 2.1. Study Design

Grant and Booth’s [34] review classification defines a ‘state-of-the-art’ review as a comprehensive search of recent literature which addresses current matters, in contrast to the combined retrospective and current approach used in other types of reviews. This offers new perspectives and highlights avenues for further research. This paper is the second in a series of ‘state-of-the-art’ reviews which examine the role of peers in substance use interventions (see [35]. for the first review focused on people experiencing homelessness). Given that the literature identified and reviewed for this systematic search exceeded what was feasible to synthesize in a single manuscript, in this paper we will focus on illicit drug overdose prevention. Peers’ roles in blood bone virus (BBV) interventions, and ‘other’ (including alcohol and tobacco) harm reduction activities, will be reported separately. 

### 2.2. Search Strategy and Selection Criteria

The initial search strategy encompassed all peer support within all types of harm reduction, and included drugs, alcohol, and tobacco. Peer support can be informal, involving ad hoc support from one individual to another, or formal, with peers trained to offer support in a structured way [35]. We applied this definition of peer support to conduct the review. As noted above, the search was later categorized into three separate groups: (1) overdose prevention, (2) BBVs, and (3) ‘other’ intervention types, due to the number of full text papers that met study inclusion criteria and to facilitate a manageable and coherent synthesis. We use the Preferred Reporting Items of Systematic Reviews and Meta-Analyses (PRISMA) guidelines [36] to report the search strategy and selection of papers.

J.M. led the development and application of the search strategies, supported by the initial review team (T.P., H.C., R.F., and K.H.). All searches were run on 26 November 2020. The Populations, Interventions, Comparators, Outcomes, and Study designs (PICOS) framework [37] was used to identify appropriate search terms and create exclusion/inclusion criteria (Table 1). Any type of article or report that mentioned all three topics of interest were considered: (1) peer support, peer workers, peer mentors, peer advocates, peer educators, peer researchers, people with lived experience; (2) substance use; and (3) harm reduction. Studies that were not eligible were missing any of the three above components. While it is good practice for systematic reviews to have inclusion/exclusion criteria, no additional inclusion/exclusion criteria were set given that a ‘state-of-the-art’ review aims to capture all potentially relevant literature published on the topic of interest and is not limited by study type [38]. An inclusive approach was therefore adopted, akin to recently published ‘state-of-the-art’ reviews [35,39]. Studies with adults or those that combined adults and youth participants were included. Studies focusing solely on the experiences of youth were excluded due to perceived differences in this group’s needs [40].

Six electronic databases (CINAHL, SocINDEX, PsycINFO, MEDLINE, Scopus, and Web of Knowledge) were searched using the search terms presented in Table 2 and adapted for each database. Searches were limited to papers published in English from 1st January 2000 and articles in peer reviewed journals. Reference details were managed using EndNote and Mendeley. The reference lists of included articles were screened for any additional papers missed by the original search strategy.

Two reviewers (J.M. and K.H.) screened for duplicates and read all titles, abstracts, and full texts of relevant articles. Relevance was assessed according to the criteria in Table 1. Discrepancies were resolved by consensus or consulting a third reviewer (R.F.). The third reviewer assessed 12 papers. Articles of any study type, including study protocols, were included. As precedented in a recent ‘state-of-the-art’ review [35], a separate grey literature search was not performed. However, relevant grey literature identified as part of the reference list screening were included in the final review.

### 2.3. Quality Assessment

Quality assessment is not used as a study inclusion criterion in ‘state-of-the-art’ reviews [34]. Papers are included based on their relevance, but methodological assessment may be conducted to increase the transparency of the synthesis and the interpretability of the findings. As reported in a recent ‘state-of-the-art’ review [35], sample sizes, data collection methods, and perceived limitations of each included paper were assessed, but no formal quality assessment tool was utilized.

### 2.4. Data Analysis 

Data concerning study design and key characteristics, including populations, interventions, outcomes, and implications for policy, were extracted by two reviewers (K.H. and J.M.) into a spreadsheet in Microsoft Excel^®^ for Windows (Washington, DC, USA). The data extraction table (Appendix A) was shared with all team members to ensure accuracy. Study characteristics, including setting, participant characteristics, and methods, were included. As this ‘state-of-the-art’ review includes any study design, as well as protocols, commentaries, and reviews, pooling the data was not possible, and a narrative synthesis was the most appropriate approach to data analysis, as informed by Snilstveit et al. [38].

An initial thematic analysis approach, led by J.M. and supported by H.C., R.F., F.M., and T.P., was used to compare papers and provide a synthesis of key points. We used Boyatzis’ [41] definition of a thematic analysis as “involving the encoding of qualitative information through the assignment of manifest and/or latent categorizations” (p. 7), with the purpose of ensuring meaningful data reduction. The analytical process was guided by Braun and Clarke’s [42] six-phase framework for conducting a thematic analysis, but this was adapted for the purposes of the ‘state-of-the-art’ review [35]. F.M. and B.P. further iteratively refined the initial thematic conceptualization via the writing/ rewriting process for this paper.

### 2.5. Identification of Relevant Papers

In total, 2020 papers were identified, 838 duplicates were removed, and 1182 publications were screened against the inclusion criteria; 196 were assessed at full text, of which 52 were excluded (see Figure 1). In total, 144 papers were identified that considered the delivery of peer support in substance use (drugs, alcohol, and tobacco) harm reduction. As described above, a total of 46 studies were included in the current review with a focus on the role of peers specifically within illicit drugs overdose prevention (no studies explored overdose prevention relative to alcohol or tobacco). All papers that met study inclusion criteria were categorized into intervention type to aid the synthesis of results. The literature searching and screening process are shown using a PRISMA flow diagram (Figure 1).

### 2.6. Characteristics of Included Papers

The 46 included papers were published between 2006 and 2020 and consisted of 12 quantitative studies [44,45,46,47,48,49,50,51,52,53,54,55]; 25 qualitative studies [15,17,23,32,33,56,57,58,59,60,61,62,63,64,65,66,67,68,69,70,71,72,73,74,75]; three mixed method studies [76,77,78]; four reviews [35,79,80,81]; one commentary [82]; and one case study [83]. Sample sizes in the 42 primary studies ranged from *n* = 6 [33] to *n* = 1392 [45]. Data collection methods varied and included: semi-structured interviews with peers and stakeholders; focus groups; story boards; ethnography; narrative accounts; secondary analysis of randomized controlled trials (RCTs); and cross-sectional designs to quantify frequency of client engagements. Most of the primary studies were conducted in North America (Canada, *n* = 20, and USA, *n* = 14 (with an additional study recruiting across both the USA and Canada [77])); with the other primary studies being conducted in the United Kingdom (*n* = 3), Australia (*n* = 1), China (*n* = 1), Italy (*n* = 1)), and the Netherlands (*n* = 1). Both grassroots and institutionalized interventions were included, mirroring the wide variety of overdose interventions involving or led by peers. Limitations acknowledged by the primary study authors included small sample sizes (e.g., [59], with eight participants, and [33], with six participants), and a lack of gender and overall diversity representation. 

### 2.7. Overview of the Included Reviews: Primary Focus 

Included papers were diverse in terms of their primary focus and the interventions they encompassed, as detailed in Table 3. The most common intervention was naloxone-related interventions. This included take-home naloxone (THN), naloxone education and training, and an exploration of barriers and facilitators for the use of naloxone amongst peers.

## 3. Results

The findings outlined below capture overdose prevention interventions, inclusive of peer support, peer-led, and peer-involved services. The outcomes and impacts of peer-involved overdose interventions for both PWUD accessing services and the peers involved in delivering those services are described. Findings are organized by intervention type: naloxone-related; safer environments for drug use; overdose prevention education; OST; drug checking; and social networks and support interventions. Within intervention types, key themes and sub-themes are identified.

Papers which are reviews, or have multi-component interventions, may be reported in more than one section if appropriate. Due to the overdose prevention focus of this paper, there were considerably more papers in some intervention types than in others, for example, naloxone-related interventions compared to OST. The imbalance across different peer-involved interventions also reflects the changing landscape of harm reduction, with newer approaches, such as drug checking, having fewer papers in the published literature. The intervention types included in this review are shown in Figure 2, and themes and subthemes within each intervention types are shown in Table 4.

### 3.1. Naloxone-Related Interventions

The largest evidence base pertained to naloxone-related interventions [15,17,33,44,46,47,48,49,52,53,56,58,59,61,62,64,68,70,72,76,77,80,82] and five themes were developed from reviewing these papers (see Table 4).

#### 3.1.1. PWUD Engaged and Committed to Peer-Involved Naloxone Interventions

Overall, PWUD were motivated and willing to administer naloxone during an overdose. To better understand THN administration, McAuley et al. [80] conducted a systematic review and descriptive meta-analysis of THN programs for people who use opioids. McAuley et al. [80] calculated that, within a three-month period following supply, around 9% of distributed THN kits were used for peer administration for every 100 PWUD who had received training. These findings contribute to evidence supporting the importance of peer administration of naloxone as a mechanism to address the overdose crisis [80]. Additionally, Bartlett et al. [68] identified eighty percent of participants as willing to administer naloxone to their peers if they had the opportunity. These authors also identified peer-driven overdose prevention as being a viable way to reach PWUD who are hesitant or unable to seek professional medical attention [68].

Galea et al. [52] and Piper et al. [46] explored the provision of naloxone to people who inject drugs as an overdose prevention strategy in New York City, USA, and identified that participants were comfortable with administering naloxone to their peers. In all instances of peer naloxone intervention, the individuals survived [52], and providing people who inject drugs with naloxone to intervene was recognized to be a useful component of a comprehensive overdose prevention strategy [46,52]. Additionally, Winhusen et al. [49] evaluated the impact of a peer recovery support service for PWUD who reported an overdose in the prior six months. Sixty-five percent of participants reported naloxone utilization, and self-reported opioid use and overdose risk behaviours were significantly decreased at each follow up [55]. Finally, in their evaluation of a peer-led non-fatal opioid overdose response, Welch et al. [48] reported high engagement with the naloxone intervention, particularly for individuals who had not previously been engaged in a naloxone distribution network. Welch et al. [48] identified the peer-led programme as a novel and replicable response to the overdose epidemic.

Peers were essential in effective delivery and knowledge transference in naloxone education programs. Gaston et al. [53] investigated the possibility of decreasing DRDs by training PWUD to recognize and manage overdoses. The authors identified most PWUD as having good retention of overdose recognition knowledge, suggesting a commitment to the process of peer education and intervention [53]. Further, Hanson et al. [56] conducted interviews with individuals with lived/living experience of opioid use who had administered naloxone to a peer during an overdose in Alaska, USA. Participants perceived naloxone to be effective and were satisfied with naloxone training, emphasizing the need to make it widely available across communities. Additionally, Winhusen et al. [49] conducted a secondary analysis of an RCT of a peer recovery support service for PWUD who had reported an overdose in the prior six months. The intervention included education regarding personal overdose risk factors and had positive implications, including a significant increase in the knowledge of overdose. Finally, Wagner et al. [47] assessed an overdose prevention and response training programme which included naloxone administration for people who inject drugs. Following the intervention, significant increases were reported in overdose understanding, largely related to naloxone knowledge.

#### 3.1.2. Benefits for the Naloxone Administrator 

Naloxone intervention has obvious positive implications for the recipient, but there are also benefits for those administering the medication, including empowerment and reduction in overdose risk. Mitchell et al. [61] explored perceptions of a THN programme in Vancouver, Canada. Researchers were young adults who had lived experience of THN, and participants were young adults (19–25 years) with mental health or substance use problems who were also experiencing homelessness. Findings outlined a sense of altruism towards peers, family, and community, as reasons to carry naloxone. A sense of empowerment from being skilled in naloxone intervention was also identified, and participants reported that their peers valued and respected this skill. Positive benefits from naloxone training were also found by Marshall et al. [33], who interviewed peer trainers with lived or living experience of drug use in THN programs. Naloxone training and intervention reduced DRDs and provided peer trainers with psychological benefits, such as self-esteem, empowerment, power in decision making, and feelings of responsibility. These benefits impacted peers’ confidence to respond in the event of an overdose and contributed to peers’ recovery [33]. In their peer overdose intervention, Wagner et al. [47] also noted unforeseen benefits, such as over half the participants reporting decreased drug use at three month follow up.

#### 3.1.3. Naloxone and Expanding Community Level Harm Reduction 

Naloxone interventions were frequently used in the context of reducing community harm through reach and integration. In their scoping review of community plans to reduce opioid-related harms, Leece et al. [77] identified naloxone education and training as being a common community strategy, and harm reduction was often framed in relation to increasing the accessibility of naloxone. The use of naloxone as a community level harm reduction strategy was also supported by Bennett et al. [44], who emphasized the importance of equipping PWUD with the skills to identify and respond to an overdose in the community. The authors identified the success of peer delivered naloxone in response to an overdose in the community, with 96% of overdoses successfully reversed following peer administration. Additionally, peer-involved naloxone interventions were found to reach communities who previously had not engaged with other harm reduction provisions. Owczarzak et al. [62] identified a peer- and street-based naloxone distribution programme as being successful in expanding harm reduction in a community where such intervention had previously been unwelcome. Naloxone was integrated into the community by challenging stereotypes, empowering marginalized groups, and deriving credibility from peers’ own community background and experiences [62]. Furthermore, Waye et al. [45] examined an Emergency Department (ED) outreach-based peer recovery service, called AnchorED, in Rhode Island, USA. Peer recovery specialists provided naloxone training and kits in the ED to patients at high risk of overdose. AnchorED was closely linked with AnchorMORE and peer recovery services in the Anchor Recovery Community Centre. Naloxone kits were distributed by AnchorMORE to high-risk communities, and peer recovery specialists partnered with resources such as local shelters, medication providers, and needle exchange programs in the community. AnchorMORE also partnered with local businesses to train staff in naloxone intervention [45].

#### 3.1.4. Barriers to Naloxone Use

Whilst findings propose PWUD as readily engaging with naloxone education, both micro and macro barriers to use of naloxone were identified. Micro barriers identified by Gaston et al. [53], included the reality that participants did not consistently carry their naloxone, making it unavailable in the event of an overdose. Issues such as the bulkiness of the naloxone had a negative impact on an individual’s desire to carry it [53]. Holloway et al. [15] also found that peers were unable to intervene due to not carrying naloxone, and in some cases lacked confidence in their ability to administer it. Additionally, Dechman [72] identified inconsistent or unreliable naloxone availability as a barrier to naloxone intervention during an overdose. 

Hesitancy to intervene during overdose with naloxone was also linked to macro factors. Holloway et al. [15] outlined factors including: drug laws; the social setting of injecting drug use; peer group drug use norms; and difficulties identifying an overdose as impeding on naloxone intervention. Additionally, Gaston et al. [53] illustrated how perceived stigma and fear of police engagement impacted on the ability and desire of individuals to carry naloxone. Kolla and Strike [58] explored findings from an overdose education and naloxone distribution (OEND) programme in Canada. OEND programs advise not to inject alone, to carry naloxone, and to call emergency services in the event of an overdose. However, the ability of peers to follow guidelines was impeded by structural vulnerabilities, such as fear of eviction if emergency services had to be called during an overdose, and criminalization of drug use [58].

#### 3.1.5. Implications of Naloxone Intervention

The implications of naloxone intervention were identified, with two main subthemes discerned: the unpleasant effects of withdrawal, and recognizing the pleasure gained from illicit drug use. Naloxone saves lives through overdose reversal. However, the potentially unpleasurable experiences of opioid withdrawal are less commonly discussed in the literature. Holloway et al. [15] explored peer responses to fatal and non-fatal overdoses and found that participants were often hesitant to intervene with naloxone due to the potential for post-resuscitation distress, or having previously witnessed a negative reaction following naloxone intervention. Parkin et al. [64] supported these findings and outlined that there were psychological reactions to withdrawal (e.g., aggression), as well as physiological reactions, which must be considered when supporting peers to intervene with naloxone. Parkin et al. [64] concluded that overdose training should be strengthened by including awareness and training for peers in how best to respond to potentially negative psychological reactions from overdose reversal. Further, Farrugia et al. [70] highlighted naloxone’s capacity to stimulate distressing withdrawal symptoms with the possibility of conflict negatively affecting the willingness of peers to use naloxone. The authors discussed the need for training to identify administration strategies, such as titration of naloxone, which are less likely to lead to a negative outcome, in order for THN uptake to increase and THN reputation to improve [70]. Mirroring these conclusions, Kolla and Strike [58] also recognized that there are a multitude of experiences following naloxone administration, not all of which are positive, and proposed that PWUD must be properly supported to deal with overdose-related stress and loss.

Involving peers in harm reduction interventions facilitates the acknowledgement of pleasure associated with drug use and reduces undue naloxone intervention. Watson et al. [82] explored naloxone use in an OPS in Canada, which largely consisted of people with lived experience of drug use. The OPS prioritised oxygen over naloxone, unless breathing was absent. Medical volunteers were paired with volunteers with lived experience of using drugs to ensure that overly medicalized perspectives of drug consumption did not dominate. This allowed for overdose prevention when required, but pleasurable drug experiences otherwise [82].

### 3.2. Safer Environments for Drug Use

Thirteen papers explored what we have termed ‘safer environments for drug use’. These are spaces such as supervised or safer consumption rooms, safer injection spaces, safer consumption sites (SCS), drug consumption rooms, supervised injection facilities (SIF), and overdose prevention sites (OPS) [23,32,35,50,55,60,66,71,76,79,81,82,83]. Safer environments for drug use differ greatly in terms of formality, bureaucracy, and peer involvement. For example, compared to SCS, OPS tend to be a lower barrier, less regulated, have fewer guidelines to comply with, and be staffed and run by peers [79,85,86].

#### 3.2.1. The Role of Peers in the Social Dynamics of Safer Environments for Drug Use 

The role of peers in connecting with individuals due to their unique identity and role is critical to provision of safer spaces. In communities underserved by safer environments for drug use, Mitra et al. [55] recognized involving peers in the delivery of harm reduction services, specifically in outreach to PWUD, as being useful in reaching those who were injecting drugs in public, women, and those who injected alone. Small et al. [66] also recognized that peer-led networks were able to meet the needs of populations who are vulnerable and not reached by more traditional facilities [66]. Similarly, Bouchard et al. [50], who conducted a peer driven social network analysis of people who inject drugs in a sanctioned SIF found that peers provided harm reduction to individuals on the periphery of the network who were marginalized or faced public stigma.

Peers also played an essential role in the mitigation of barriers and risks for PWUD. In their ethnographic study of a peer-run SCS in Vancouver, McNeil et al. [60] acknowledged the disproportionate vulnerability to drug-related harms that people who require help injecting were exposed to. Peers providing injecting assistance helped individuals regain control and autonomy over their consumption of drugs, as individuals were no longer reliant on others when injecting [60]. Additionally, in terms of specific vulnerable populations such as people experiencing homelessness, Bardwell et al. [71] suggested that the implementation of a peer-based supervised injection intervention in emergency shelters could positively influence injecting drug use and overdose risks [71]. Finally, McNeil and Small [81] found mobile and peer-based interventions to be successful in disrupting inequities that shape social and spatial drug environments, such as relief from exploitive relations with “‘hit doctors’ (i.e., someone who provides assistance with injections)” (p. 496).

Although safer environment interventions are effective in mitigating some risks associated with drug use, they are limited by micro and macro constraints. Bardwell et al. [71] reported that, despite peer support reducing stigma and shame through the normalization of drug use in shelter and hostel spaces, factors which limit intervention effectiveness are rooted in underlying social norms and material constraints, meaning people continue to inject alone in non-designated injecting spaces [71]. McNeil et al. [60] also recognized constraints to SCS, stressing that systemic barriers to accessing services exist. The authors identified that other peer-led resources, such as assisted injecting, are required to increase the reach of harm reduction [60].

#### 3.2.2. A Trusted Environment Created by Peers 

In conducting this review, the importance of safety, connection, and trust emerged as critical to facilitating safer environments for PWUD. Bergamo et al. [76] reported findings from one unsanctioned peer run SIF which provided PWUD with 24/7 naloxone availability. In this peer-run facility, not a single overdose death was recorded in the 10 years the service was running [76]. The authors provided further support for the de-medicalization of overdose prevention using naloxone, and future integration of peer-based models in harm reduction [76]. Impacts of increased safety, such as no overdose deaths, were also found by Pauly et al. [23]. These authors outlined some of the impacts of overdose prevention sites involving peers, including zero deaths, increasing trust, earlier intervention to prevent overdoses, and development of relationships, reduced trauma, and the mitigation of stigma [23]. 

In their research on peer worker involvement in low-threshold supervised consumption facilities in Vancouver, Canada, Kennedy et al. [32] reported that peer-involved safer environments for drug use facilitated harm reduction practices and promoted health benefits. Additionally, Small et al. [66] identified that their peer-led injecting support team within Vancouver Area Network of Drug Users (VANDU) provided peer-to-peer education and assistance during the injecting, rather than after or before. This novel approach responded to gaps in harm reduction related to unsafe injecting practices [66]. Miler et al. [35] also supported recommendations for the inclusion of people with lived experience within high-risk environments, such as safer environments for drug use. In their ‘state-of-the-art’ review of peer interventions within homelessness and substance use settings, Miler et al. [35] illustrated the importance of peer-involved environments in minimizing overdose risk. 

In addition to increasing feelings of safety for PWUD, peer involvement in safer environments for drug use also increased feelings of comfort. Kennedy et al. [32] concluded that peer involvement was characterized by feelings of comfort, with the lived experience and expertise of peers being particularly important in facilitating this. PWUD also felt more comfortable with peers due to feeling less rushed by them than by health professionals. The authors also outlined that increased levels of comfort positively impacted on an increase in engagement with safer environment services [32]. Additionally, Pauly et al. [23] noted that peer-to-peer support increased comfort for individuals using OPS.

The involvement of peer workers in safer environments for drug use helped build feelings of trust between those using the services and those providing them. In their review of OPS, Pauly et al. [23] outlined the pivotal role which peers had in increasing feelings of trust for PWUD. Safer environments for drug use were outlined by the authors as facilitating a shift from shame and blame to trust and the development of relationships. Trust was enhanced through spending time together and not having to conceal drug use. The enactment of trust was essential to build an effective, accepted service for PWUD [23]. Additionally, Kennedy et al. [32] identified that trust was the mechanism through which PWUD and peers with lived experience created feelings of comfort. In their review of peer support at the intersection of homelessness and substance use, Miler et al. [35] also outlined peers as creating strong, trusting, experience-based relationships with service users, and concluded that the unique position of peers was an important factor in their ability to gain trust. Small et al. [66] mirrored these findings and proposed that individuals with lived experience in their injection support team were trusted by their peers and the community. 

#### 3.2.3. Benefits for Peer Workers in Safer Environments for Drug Use 

Peer workers identified that a sense of purpose, being an inspiration for others, and a sense of belonging, motivated them to work in safer environments for drug use, despite the challenges. Pauly et al. [65] identified that, even with frequent hardships and loss, peer workers found their work to be meaningful, and they identified potential strategies to buffer some of the associated stress and trauma. Kennedy et al. [32] also outlined benefits for PWUD volunteering at OPS, such as improving quality of life, expanding skillsets, and developing résumés.

#### 3.2.4. Challenges for Peer Workers in Safer Environments for Drug Use

Although safer environments for drug use were beneficial for both PWUD and the peers who provided them, significant challenges for peer workers were also identified. Challenges at the individual level, in addition to organizational and macro challenges, were recognized, and these included stress, trauma, lack of recognition, and inequitable pay. Pauly et al. [65] identified that working in safer environments for drug use is stressful, with lasting emotional and mental health effects. Even so, the authors acknowledged the dedication of peers to working in such environments, with peers desiring to provide support and safe environments for service users [65]. Although central to the delivery of safe environments for drug use, Pauly et al. [23], Pauly et al. [65], and Kennedy et al. [32] reported that peer workers were often unfairly compensated, and the value of their work was unrecognized or their roles misunderstood. Whilst peers were integral in harm reduction within safer environments, Pauly et al. [23] noted limited funding, limited physical space, and dangers of criminalization, as additional stressors for workers [23]. In addition to minimal financial compensation, Kennedy et al. [32] and Pauly et al. [65] identified that many peer workers received minimal support in their role, with Kennedy et al. [32] specifically outlining a lack of support for considerable grief during the overdose crisis.

Changes to institutionalization and formalization of services represent further challenges for peer workers to navigate. Watson et al. [82] conducted a critical review of Canadian harm reduction services and acknowledged the lifesaving capacity of safer environments for drug use. However, the authors identified that, when services become formally bureaucratized, they are less innovative, dynamic, and responsive to changing drug landscapes, as well as being less inclusive of people with lived experience [82]. The impact of political climate and bureaucratization was also echoed by McNeil and Small [81] in their review of safer environments for drug use. These authors identified that, whilst these spaces mitigated drug-related harms and increased access to social and material resources, such interventions were constrained in many countries by drug laws and law enforcement activities [81]. Additionally, Kerr et al. [79] reviewed the history of safer environments for drug use in Canada from 1994 to 2017 and reported that, despite considerable evidence for the successes of these environments in reducing the health and social harms for PWUD, extraordinary efforts and activism by PWUD and healthcare providers are required to preserve them. 

### 3.3. Overdose Prevention Education 

Seven papers explored overdose prevention and harm reduction education initiatives [47,53,57,58,66,75,77]. Three themes were identified including: (1) peers providing actionable knowledge and skills; (2) impact of overdose prevention education on behaviour; and (3) limitations of peer inclusion in overdose prevention education.

Peers provided PWUD with both general knowledge and specific harm reduction education and skills. Two papers explored education provided by peer-run organization VANDU [57,66]. Kerr et al. [57] outlined several education and support programs for PWUD, including peer mentorship. Education groups provided knowledge of social and health issues, in addition to peer support, which contributed to direct action to address specific inequities faced by PWUD in Vancouver, Canada. Additionally, Small et al. [66] discussed a specific project from VANDU which provided safer injecting education. The VANDU injecting support team provided guidance and instruction which could be actioned to reduce harm immediately. Peer-to-peer injecting education was identified as being essential for PWUD who lacked knowledge of injecting techniques. Peer educators reduced multiple forms of injecting-related risk, such as getting a ‘clean shot’, in addition to reducing human immunodeficiency virus (HIV) and Hepatitis C virus (HCV) risk [66]. Peers were also essential in conveying information through formats such as peer modelling. Green et al. [75] systematically reviewed educational overdose videos for justice-involved individuals and outlined gaps in available material. To address these, focus groups with people who used opioids and formerly justice-involved individuals guided discussions on material which should be contained in educational videos. A ‘peer trainer’ and ‘peer learner’ format was preferred as individuals liked seeing and hearing from people with lived or living experience of substance use, with many corroborating their own experiences with the peers on the videos [75].

Overdose prevention education was recognized as having implications for overdose intervention and other behaviours, such as a reduction in drug consumption. Kolla and Strike [58] explored implementation of one OEND programme and reported that the specific health education for PWUD contributed to their ability to provide care to their peers during overdose events. The authors reported that OEND programs acknowledged that PWUD are often the first line of response in the event of an overdose, supporting the idea of reciprocal aid and harm reduction strategies within social networks [58]. Wagner et al. [47] also reported improved knowledge and overdose response behaviour among people who inject drugs who attended training. Of those who witnessed an overdose during the training follow-up period, around 90% actively intervened. Furthermore, around half of those overdosing were strangers, suggesting that overdose training and education may have positive implications for the wider community [47].

In the findings, we found peer-involved education programs to have good retention across follow-up periods [53]. However, there were limitations related to the integration of peers in overdose prevention programs. On evaluation of community-based projects to reduce DRDs, Leece et al. [77] identified three critical omissions within community strategy. These included a lack of inclusion of people with lived experience, community-led approaches and peer-driven approaches [77].

### 3.4. Opioid Substitution Treatment (OST)

Four papers explored OST [17,54,69,78], with three themes identified: (1) low-barrier community reintegration; (2) peers as experts; and (3) challenges associated with OST. OST can be an initial point of contact for people reintegrating into the community. 

Krawczyk et al. [54] evaluated a low threshold buprenorphine outreach programme with peer recovery coaches to engage populations who are justice-involved and largely disconnected from treatment. The authors identified peer recovery specialists as working with individuals to help them re-engage with the community, acquiring ID (identification papers) and insurance, and engaging with housing programs, noting that such linkages were key components of the outreach programme [54]. Peer staff also recognized their own ability to assess and understand drug-related risk. Olding et al. [78] evaluated a low-barrier, peer-staffed, comprehensive community-based harm reduction site in Vancouver which included drug checking, an OST programme, and an OPS. Peer staff identified that, in addition to their training, their lived experience of substance use allowed for them to confidently assess individual tolerances and overdose risk [78]. PWUD also appreciated the education and knowledge they received from peers, and Boucher et al. [17] outlined that peers were strongly valued in OST, with participants often indicating that they learned most about harm reduction from peers [17]. However, Boucher et al. [17] identified inflexible service delivery in OST, including difficulty in receiving or maintaining ‘take home’ dosages, and restrictions when trying to change dosage, as being a primary concern for PWUD. In terms of the provision of education, Boyd et al. [69] presented experiences of a peer-run group that formed following Canada’s first heroin assisted treatment (HAT) trial. The group educated peers, supported potential participants through the research process, and advocated for change to improve the lives of PWUD [69]. 

### 3.5. Drug Checking

There was limited evidence regarding peers’ roles and responsibilities in drug checking services for people with problem drug use. Four papers examined drug checking interventions [63,74,78,84], with two themes identified: (1) hesitancy to engage with technology; and (2) social checking.

Drug checking technology as a harm reduction intervention involving peers is developing but concerns regarding responsibility and liability of the service provider have impeded engagement with technology. Olding et al. [78] evaluated a peer-staffed SCS in Canada which also offered drug checking. The authors indicated the benefits of drug checking, such as preventing overdose deaths. However, the specific role of peers in the service was ambiguous [78]. When investigating stakeholders’ perspectives on drug checking technologies and implementation, Glick et al. [74] identified stakeholders (including peer groups) as responding positively to drug checking technology. Nonetheless, the authors reported apprehensions regarding service implementation, including issues with trust and rapport between providers and PWUD, legality and policy concerns, and technological accuracy [74]. Hesitation related to legal implications was also outlined by Palamar et al. [63] who noted barriers to drug checking, such as fear of arrest, insurance liability, or legal repercussions. Similarly, reluctance to engage with drug checking services for fear of prosecution, surveillance, or police presence was outlined by Wallace et al. [84]. The authors highlighted criminalization of PWUD and/or selling drugs as a concern for prospective service users, and there was a hesitancy to engage with drug checking services for fear that police would have access to information. Wallace et al. [84] suggested that engaging people with lived experience in the drug checking service would facilitate trust with service users, and proposed trust as being essential for successful implementation of drug checking services.

Palamar et al. [63] identified PWUD as being committed to improving drug safety for themselves and others, especially in the absence of adequate government policies. These authors identified a type of drug checking known as ‘social checking’. Distinct from formal drug checking services, Palamar et al. [63] explored the rationale for individual drug checkers across North America. Testing other people’s drugs appeared to be motivated by altruism and the desire to increase safety and minimize the risks associated with drug consumption. 

### 3.6. Social Networks and Support Interventions

Three papers discussed peer-involved harm reduction interventions which focused on the creation and development of social networks and social support [51,57,67]. A dominant theme of reciprocity was identified.

To assess the available forms of social support for PWUDs, Elkhalifa et al. [51] conducted a social network analysis related to harm reduction provisions. The authors identified that the likelihood of assisting during an overdose was associated with an individual having someone in their network who provided them with tangible support, including food, money, drugs, or administering naloxone. Elkhalifa et al. [51] noted that just one source of tangible support increased the likelihood of the individual providing peer support themselves [51]. Reciprocity in support provided was also identified in virtual communities by Van Schipstal et al. [67]. The authors discussed the harm reduction strategies used in the online communities to reduce the risk of psychoactive drugs, including PWUD reaching out to their peers to practice informed and careful drug use, and peer-to-peer practical advice such as how best to measure specific substances [67].

Multifaceted support was offered by peers to PWUD, and the support provided was often beyond expectations related to professional roles. To explore the kinds of support provided by peers in a health setting, Kerr et al. [57] evaluated a ‘hospital programme’ implemented by a Vancouver peer-led organization VANDU. This programme was created to address some of the difficulties PWUD face in hospital which can lead to self-discharge against medical advice. Volunteers visited their hospitalized peers regularly, encouraging them to stay in hospital and talking to them about what was happening in their community. In addition to providing health support, volunteers also provided their hospitalized peers with social support and encouragement [57].

## 4. Discussion

There is an emerging interest in the role of peers in overdose prevention, with 32 of the 46 included papers being published from 2017 onwards. Integrating peers in services for PWUD increases their accessibility and acceptability [24] but, to date, there has been little synthesis of the published literature on the role of peers in overdose prevention for PWUD. Studies varied in their focus, design, and methodology, contributing to rich and detailed findings.

All papers identified positive impacts of peer-involved interventions in overdose settings for PWUD. Benefits were identified for both PWUD who engaged with interventions and the peers who provided them. Positive impacts included: feelings of trust, safety, and comfort in services (e.g., [23,32,76]); a reduction in DRDs (e.g., [78]); increased engagement with services through referrals (e.g., [45]); increased naloxone knowledge (e.g., [73]); psychological benefits for the peer worker (e.g., [33]); and a reduction in harms associated with injection practices (e.g., [60]). It is, however, important to acknowledge the wider constraints that harm reduction interventions operate within which are related to barriers to engagement for PWUD. These include the criminalization of drug use (e.g., [23]), fear of harassment (e.g., [81]), and entrenched and pervasive social stigma (e.g., [87]).

In the findings, we found that, in addition to their training, peers’ lived experience of substance use allowed them to confidently assess individual tolerances and overdose risk [78]. However, recent findings from Mamdani et al. [24] found that having lived experience, or being an ‘expert by experience’, was felt by peer workers to be less valued by others compared to traditional education. To bridge this gap, Mamdani et al. [24] propose that organizations should recognize and value peer workers, provide organizational supports and opportunities for training in technical skills such as CPR, in addition to the development of personal self-care skills, such as mindfulness and meditation. Mirroring findings from the review [23,25], Mamdani et al. [24] identifies the stress associated with working in overdose prevention and proposes that this stress has been compounded by COVID-19, with peer workers being disproportionately negatively impacted by the pandemic in addition to the opioid crisis. The requirement for organizations to provide training and support for peer workers that is tailored to the realities of having lived or living experience of substance use is therefore essential [24].

This review concludes that the full impact and effectiveness of peers in overdose interventions cannot be realized due to organizational and institutional barriers (e.g., [88]). Additionally, ambiguous peer roles and a lack of organizational understanding or value of peers [51,62,78] are substantive barriers to peer involvement in overdose intervention contexts. Similar findings are reported by Greer et al. [22], who emphasise the need for improvements in organizational understandings of peer roles, in addition to enhanced clarity of roles. Additionally, Greer et al. [22] stress that change is required to: (1) legitimize the contribution of peer workers in harm reduction settings; and (2) acknowledge the structural constraints which negatively impact on their experiences within the workplace. Findings from Mamdani et al. [24] also echo this, with authors proposing that systemic barriers in the working environments of peers must be acknowledged if effective interventions are to be designed which mitigate the stressors peer workers face in overdose response settings. Mamdani et al. [24] also highlight implications from the shift from grassroots to institutionalized harm reduction, which raise challenges for peers, both at micro- and macro-levels.

Related to stressful or oppressive working environments, high levels of burnout were identified in peer workers in our review(e.g., [32]). This finding was also noted by Olding et al. [78], who report that, in addition to systemic barriers, task ‘shifting’ (where shifting of overdose response tasks to less specialized workers occurred) contributed to peer worker burnout. Olding et al. [78] suggest that the devaluing and casualization of overdose response labour compounded structural inequalities for peer workers and led to emotional exhaustion and burnout. Interventions to address peer worker burnout must therefore extend beyond the provision of services at the individual level and include changes in organizational working conditions, as previously mentioned by Greer et al. [22] and Mamdani et al. [89]. In addition to stressful working environments, peers are often unfairly remunerated [23,32] and may have limited or no access to benefits or services which are available to salaried employees [23,32]. Recent literature highlights the limited opportunities peers have to recharge [88], de-stress, or participate in self-care [89]. Peer workers at the forefront of the overdose response identify that they are running themselves “ragged” [24].

### 4.1. Implications for Policy, Practice, and Research 

Peers play an essential role in activism, especially in securing funding for harm reduction facilities [35,69,79]. Sustainable funding, however, remains a considerable barrier to the provision, accessibility, and effectiveness of overdose prevention services [23]. The precarious nature of peer workers in harm reduction services is outlined in this review, and recent findings from Greer et al. [90] note that funding cuts in the non-profit sector also impact on the funding available to compensate staff. This means peer workers are often unpaid or poorly paid, and services are heavily reliant on unpaid volunteers [90]. Harm reduction services must feel secure and stable for service users but also for peers who are providing such services [81]. Future peer-involved harm reduction interventions should engage with findings that highlight peer workers as poorly compensated [23,32,90]. Related to the requirement of fair compensation, Marshall et al. [16] emphasises the value of PWUD and lived experience expertise in harm reduction interventions.

In their systematic review into the roles of people who inject drugs in harm reduction, Marshall et al. [16] identified strategies to improve peer involvement in harm reduction initiatives, including: creating opportunities for key stakeholders to engage in harm reduction dialogue; dedicating time and organizational resources to developing positive rapport with communities, gatekeepers, the general public, and those in government; fostering organizational cultures which support meaningful participation and leadership from PWUD; providing appropriate training and supervision to peer workers; and addressing barriers to the participation of PWUD by creating support which acknowledges broader determinants of health. One recent example of an intervention to engage with these recommendations related to organizational strategies for peer-involved interventions, peer skill development, and organizational support for peers, is the ROSE intervention by Mamdani et al. [89]. This intervention was developed from key support needs identified by peer workers: “R” for recognition; “O” for organizational support; “S” for skill development’; and “E” for everyone, and it aims to create a more just and fair work environment for all workers in overdose response settings. It is important to note that the ‘ROSE’ intervention was conceptualized prior to COVID-19. This has implications for peer workers as they are disproportionately burdened by the dual health crises. Whilst the intervention needs to be evaluated and tested, the authors suggest that it has the potential to be tailored to alternative geographical settings whilst maintaining the integrity of strategies used to challenge negative and stigmatizing work environments and interactions [89].

To enable the full effectiveness of peer-involved harm reduction interventions, drug policy reform is required [58,81]. Even with Good Samaritan legislation (laws which encourage people to call emergency services during an overdose by providing immunity from selected drug arrests, such as low-level possession) [91], there is limited protection for PWUD at the scene of an overdose, especially if they are on parole or probation [58]. Limitations of legislation, in addition to structural vulnerabilities and structural violence, mean that multiple risk factors impact upon an individual’s behaviour at a micro-level, and consequently upon the effectiveness of harm reduction interventions at a macro-level. Marshall et al. [16] also mirror these findings and highlight the need to address the criminalization of PWUD, initiate anti-stigma campaigns related to drug use, and implement and evaluate interventions to address individual, organizational, and systemic barriers for PWUD involvement in harm reduction initiatives. Additionally, Marshall et al. [16] outline the importance of developing a consensus framework for describing and categorizing peers’ roles in harm reduction, something that is particularly important in terms of supporting and recognizing the value of grassroots initiatives which are central to harm reduction innovations. Furthermore, there must be an acknowledgement that institutionalized harm reduction is often a result of grassroots activism from PWUD and their peer networks.

The complexities of conducting research with grassroot organizations must also be considered. In their review of ethics of community-based research with PWUD, Souleymanov et al. [92] identified that there are several ethical challenges unique to this type of research. The emphasis placed on relationships with participants and communities, the engagement with individuals with lived experience, and use of data analysis approaches which engage members of the community with partial or full access to potentially confidential information, must be considered. This type of research therefore requires methodological flexibility to match the ethical expectations of those providing services [93]. Additionally, it is important to recognize that peer researchers can face institutional barriers which inhibit their engagement within research and potential for employment as researchers. Salazar et al. [93] identifies previous encounters with the criminal justice system and/or prior felony convictions as locking PWUD out of research. The authors outline the need for research institutions to better consider the balance between accountability and flexibility in engaging justice-involved PWUD in research [93].

Finally, future peer-involved harm reduction interventions should engage with findings that peers are experiencing high levels of stress and burnout [23,25,26,65,78]. Research must explore not only what peer workers can contribute to services but also how services can nurture and protect peer workers from the stress and burnout they experience in their roles in overdose settings. Future research must consider improving intervention design, through thoughtful reflection on peer roles and responsibilities, in addition to improving the reporting of studies so that peer involvement is more transparent and measurable. While public health responses to the overdose crisis tend to focus on individual decision making and behaviour change [94], broader cultural and organizational changes must also take place to address the systemic issues that this review has highlighted.

### 4.2. Strengths and Limitations 

Within this review, considerable steps were taken to promote accuracy, reliability, and transparency within the review process itself and in presenting the findings. All stages of searching and screening papers, data extraction, and analysis were conducted by at least two members of the team. Within ‘state-of-the-art’ reviews, no formal quality appraisal is adopted [34]; however, notes on quality and clarity were taken during the data extraction phase. Most primary research was conducted in Canada and the USA (*n* = 35) which may limit the generalizability and applicability of findings beyond a North American context, especially when comparing against countries with different justice systems, welfare entitlements, or healthcare systems. This paper systematically reviews the literature and does not provide theoretical explanations for the findings. This is partly influenced by the nature of the review but is also by the desire for findings to provide pragmatic guidance to address gaps in future research and optimize current policy and practice.

## 5. Conclusions

Peers are pivotal to the delivery of effective overdose prevention services that are acceptable and accessible to PWUD. Services which integrate peers are more likely to reach vulnerable groups and integrate into communities. Peers’ lived experience increases levels of trust, safety, and comfort for the PWUD who utilize these services. The knowledge and expertise of PWUD provides a source of actionable harm reduction information and skills. Involvement in overdose interventions can also have positive implications for the peers themselves, such as feeling empowered, contributing to skillsets, and improving quality of life. However, peers involved in overdose prevention interventions commonly have to navigate stressful and traumatic work environments, may face burnout, and are often poorly compensated. Implications of the shift from grassroots to institutionalized harm reduction must also be acknowledged, with importance placed on the innovation and responsiveness of grassroots organizations, and the challenges associated with institutionalization of harm reduction for the role of peers. Future practice must recognize and value the unique position of peers in overdose interventions by clarifying and integrating their role in organizational policy, providing adequate support and appropriate remuneration.

## Figures and Tables

**Figure 1 ijerph-18-12073-f001:**
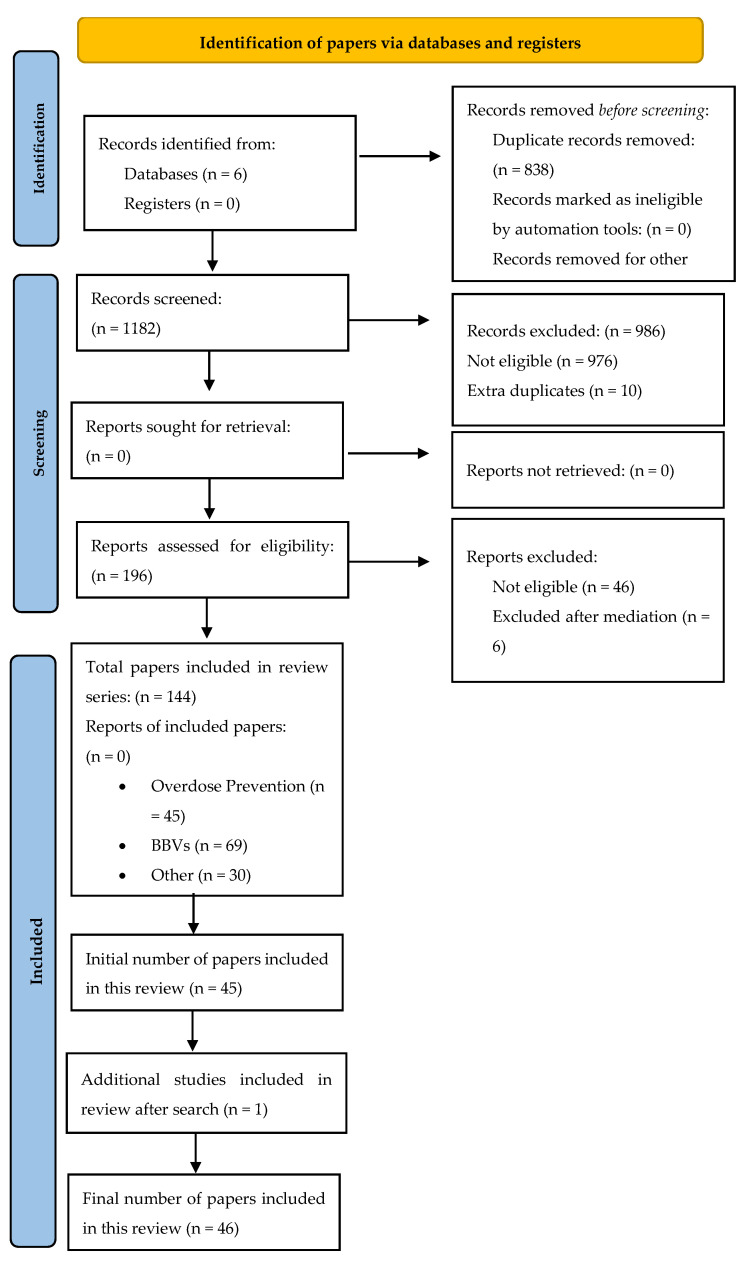
PRISMA 2020 flow diagram (adapted to include one additional study after the initial search). From [43].

**Figure 2 ijerph-18-12073-f002:**
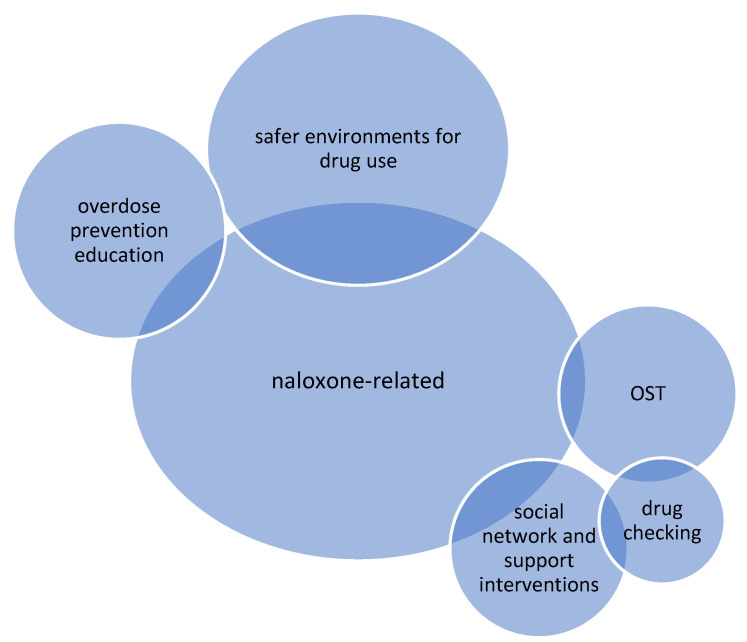
Intervention types included in this review; n.b., size is illustrative of the number of papers included in review of each intervention type. OST = Opioid substitution treatment.

**Table 1 ijerph-18-12073-t001:** Inclusion/exclusion criteria.

Inclusion	Exclusion
Populations	
People using illicit drugs (including poly-substance use—i.e., concurrent use of various substances) Drugs used both problematically and/or recreationally Adults (over 18 years, with no upper age limit)	Non-drug use Under 18s
Interventions	
Any/all types of harm reduction approaches involving peers/mentors	Non-harm reduction approaches; non-peer approaches
Comparators	
Any None (i.e., evaluations or studies focusing on harm reduction peer interventions with no comparators)	N/A
Outcomes	
Any	N/A
Study design	
Any study design	N/A

N/A = not applicable.

**Table 2 ijerph-18-12073-t002:** Search Strategy.

Operator	Definition
**PsychINFO**	
Title (TI)/Abstract (AB)/Keywords (KW): Intervention	TI (peer support OR peer intervention OR peers OR peer mentor) OR AB (peer support OR peer intervention OR peers OR peer mentor) OR KW (peer support OR peer intervention OR peers OR peer mentor)
2. Subjects (SU): Intervention	SU (peer support OR peer intervention OR peers OR peer mentoring OR lived experience)
3. Boolean Operator	1 OR 2
4. Title (TI)/Abstract (AB)/Keywords (KW): Population	TI (substance use or substance abuse or substance misuse or drug use or drug abuse or drug misuse or dependence or addiction or alcohol use or alcohol misuse or alcohol abuse or alcoholism or smoking) OR AB (substance use or substance abuse or substance misuse or drug use or drug abuse or drug misuse or dependence or addiction or alcohol use or alcohol misuse or alcohol abuse or alcoholism or smoking) OR KW (substance use or substance abuse or substance misuse or drug use or drug abuse or drug misuse or dependence or addiction or alcohol use or alcohol misuse or alcohol abuse or alcoholism or smoking)
5. Subjects (SU): Population	SU substance use or substance abuse or substance misuse or drug use or drug abuse or drug misuse or dependence or addiction or alcohol use or alcohol misuse or alcohol abuse or alcoholism or smoking
6. Boolean Operator	4 OR 5
7. Title (TI)/Abstract (AB)/Keywords (KW)/Outcome	TI harm reduction OR AB harm reduction OR KW harm reduction
8. Subjects (SU): Outcome	SU harm reduction
9. Boolean Operator	7 OR 8
10. Boolean Operator	3 AND 6 AND 9
11. Language limit	English language
12. Time limit	2000–2020
13. Selection	Removal of duplicates followed by PRISMA guidelines of article sifting: title sift, abstract sift, full-text sift, review reference lists, and articles citing.

**Table 3 ijerph-18-12073-t003:** Overview of the included papers (*n* = 46 *). (* 46 papers; some interventions were multicomponent).

Interventions	Description of Intervention	Number of Papers	Papers
Naloxone-related interventions	Naloxone is a medication that can reverse the effects of opioids to save lives during an overdose. Interventions include training in recognising signs of overdose and safely administering naloxone, take-home naloxone (THN) programmes which provide naloxone to be used when witnessing an overdose, and peer programmes where people who inject drugs are trained to use it to help their peers.	23	[15,17,33,44,46,47,48,49,52,53,56,58,59,61,62,64,68,70,72,76,77,80,82]
Safer environments for drug use	Creating environments which offer safety or supervision during drug consumption to reduce risk of overdose, such as safe consumption sites. Spaces can be staffed with paid employees, volunteers, or peers trained in the use of naloxone.	13	[23,32,35,50,55,60,66,71,76,79,81,82,83]
Overdose prevention education interventions	Provision of training to bystanders regarding how to intervene during a witnessed opioid overdose. This includes brief interventions, education, and training related to overdose management and intervention.	7	[47,53,57,58,66,75,77]
Opioid substitution treatment (OST)	Treatment to reduce drug dependence and injecting frequency by offering people who are opioid dependent an alternative, prescribed medicine, typically methadone or buprenorphine.	4	[17,54,69,78]
Drug checking	Drug checking services enable individuals to have their drugs chemically analysed, providing information on the content of the samples as well as advice, and, in some cases, counselling or brief interventions.	4	[63,74,78,84]
Social network and support interventions	Relational interventions across different contexts which focus on the provision of social support and connection to PWUD by peers.	3	[51,57,67]

**Table 4 ijerph-18-12073-t004:** Themes and subthemes.

Intervention Type	Theme	Subthemes
Naloxone-related	PWUD * engaged and committed to peer-involved interventions	Naloxone administration Naloxone education and training
	Benefits for the administrator	Empowerment Reduction in risk
	Naloxone and expanding community level harm reduction	Reach in the community Integration in the community
	Barriers to naloxone use	Micro barriers Macro barriers
	Implications of naloxone intervention	Unpleasant effects of withdrawal Pleasure from illicit drug use
Safer environments for drug use	The role of peers in the social dynamics of safer environments for drug use	Connectivity Mitigation of barriers and risk Constraints
	A trusted environment created by peers	Safety Comfort Trust
	Benefits for peer workers in safer environments for drug use	
	Challenges for peer workers in safer environments for drug use	Individual challenges Organizational and macro challenges
Overdose prevention education	Peers providing actionable knowledge and skills	
	Impact of overdose prevention education on behaviour	
	Limitations of peer inclusion in overdose prevention education	
Opioid substitution treatment (OST)	Low-barrier community reintegration	
	Peers as Experts	
	Challenges associated with OST	
Drug checking	Hesitancy to engage with technology	
	Social checking	
Social networks and support interventions	Reciprocity	

* PWUD = People Who Use Drugs.

## Data Availability

The data presented in this study are available in the Appendix A submitted to IJERPH alongside this manuscript.

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
