# Peer review of "Peer Support and Overdose Prevention Responses: A Systematic ‘State-of-the-Art’ Review"

_ijerph, 2021, doi:10.3390/ijerph182212073_

Round 1
Reviewer 1 Report
This is well-written and comprehensive review of peer support interventions, and I only have one large and multiple minor critiques.
- My major comment is that the results are incredibly detailed to the point where they can be overwhelming to the reader. It could serve to do some reducing to get to the main points by simply stating how many papers applied to the theme and then focusing on describing the theme with the most appropriate/strongest example listed, rather than listing multiple examples. The table provides a link between the themes and all of the articles, so the large amount of detail in the narrative is not necessary.
- Minor issues:
- End of fourth paragraph in the intro looks like it is cut off.
- Re-defining peer supports in the methods is redundant with the introduction.
- Please define PICOS in the text.
- OST is stated to be "low-barrier in the discussion. I am not sure what the authors consider to be low-barrier, as research has demonstrated considerable barriers to access of these interventions that prevent most people from starting or being retained on them.
- In the discussion, it might be worth pointing out how there can be difficulties conducting research with grassroots organizations, which often require methodological compromises to match with ethical perspectives of those who deliver the services.
- At the end of the discussion (second to last paragraph), it would be worth mentioning how peers can have difficulties accessing certain spaces to conduct this work, such as jails and prisons if they have felony backgrounds.
Author Response
Please see the attachment. Thank you for your revisions.

Reviewer 2 Report
The manuscript presented for review concerns the state of the art of peer support and overdose prevention responses. The article is based on systematic review of 46 papers. The paper has a clear structure and analyzes the issues raised in detail. In my opinion, it is a valuable source of knowledge and can be published.
Suggestions of minor improvements are listed below:
Introduction section Lack of dots in the expression: “Authors emphasize that, despite the com-plex and demanding nature of the work, peers often feel undervalued and ineffective due to a lack of organizational understanding about their roles and the value of experiential knowled”
- Add references to the above sentence
-The titles of the sections 3 and 4 e.g. Results and Findings are incomprehensible and inconsistent with the journal’s guidelines. In section 3 you have described methodological aspects of your study – inclusion/exclusion criteria etc. and the appropriate Results section - comprehensible with the main goal of the study is in section 4. I suggest to change the title of section 4 to Results
- The authors started the section 4 with describing methodological aspects – paragraph started with “Both grassroots…..” – this should be in the previous section and the results should concentrate only on main goal e.g.overdose prevention interventions that engaged peers.
- I don't understand the need for figure 2 - what does the difference in wheel sizes mean - is it a coincidence or a graphical presentation of the number of articles for each topic? Especially, as it is not explained in the paper
- In section 4.1.2 in the first sentence you stated “(….) reduction in risk” – however you did not add what kind of risk do you mean?
Author Response

(The authors gave the same response as above.)

Reviewer 3 Report
Very interesting article. At first I thought its length would be tiresome both for me as a reviewer and for the readers. However, the way of narration and at the same time meticulously described research process proved to be very interesting and despite its length engaging. The material that was compiled could be successfully divided into two publications (e.g. one on Naloxone-related interventions), however, on reflection, I admit that those who are patient and interested in this topic will find the article very interesting.
Main strengths: clear methodology and rich analyses.
Minor corrections to be made: removing the blank page 6 (including the title "Figure....").
It also seems to me that Figure 1 is redundant. The material selection process is sufficiently described. However, I leave the decision to the authors.
Please also adjust the References section according to the guidelines (including numbering, punctuation for surnames and first name initials, periods in journals abbreviations, italics in the volume number, spacing, unnecessary bold in the last reference, etc.).
Author Response

(The authors gave the same response as above.)
